# A Structural Perspective on the RNA Editing of Plant Respiratory Complexes

**DOI:** 10.3390/ijms23020684

**Published:** 2022-01-08

**Authors:** Maria Maldonado, Kaitlyn Madison Abe, James Anthony Letts

**Affiliations:** Department of Molecular and Cellular Biology, University of California, Davis, CA 95616, USA; kamabe@ucdavis.edu

**Keywords:** plant respiration, plant mitochondria, RNA editing, structure-function

## Abstract

The last steps of respiration, a core energy-harvesting process, are carried out by a chain of multi-subunit complexes in the inner mitochondrial membrane. Several essential subunits of the respiratory complexes are RNA-edited in plants, frequently leading to changes in the encoded amino acids. While the impact of RNA editing is clear at the sequence and phenotypic levels, the underlying biochemical explanations for these effects have remained obscure. Here, we used the structures of plant respiratory complex I, complex III_2_ and complex IV to analyze the impact of the amino acid changes of RNA editing in terms of their location and biochemical features. Through specific examples, we demonstrate how the structural information can explain the phenotypes of RNA-editing mutants. This work shows how the structural perspective can bridge the gap between sequence and phenotype and provides a framework for the continued analysis of RNA-editing mutants in plant mitochondria and, by extension, in chloroplasts.

## 1. Introduction

RNA editing introduces alterations into coding or non-coding regions of RNA molecules after transcription. Alterations include insertions, deletions and changes to the identity of the genomically encoded ribonucleotide, affecting the molecule’s stability, splicing and binding to regulatory factors, as well as the encoded protein sequence in the case of messenger RNA. Thus, RNA editing has significant consequences on an organism’s gene expression.

RNA editing may occur in the nucleus, cytoplasm, or DNA-containing organelles. In land plants, extensive RNA editing occurs in mitochondria and chloroplasts [1,2,3,4]. This mechanism, which is thought to have originated 500 million years ago upon land colonization, is widespread across all major terrestrial plant clades [5]. More than 700 RNA-editing sites have been identified in multiple mitochondrially encoded proteins, including complexes of the respiratory chain, ribosome subunits and mitochondrial transporters [6]. Similarly, over 40 sites have been identified in proteins encoded in the chloroplast [5]. Plant organelle RNA editing mostly involves deamination of cytidine to uridine (C-to-U conversion) through the activity of over 400 pentatricopeptide repeat (PPR) proteins and associated co-factors [7]. As most changes occur in the first and second nucleotide of the codon (~30% and ~55%, respectively), the edits generally change the encoded amino acid [8]. Additionally, edits may create start codons, stop codons and splicing sites [6]. Given that the sites are in essential subunits of the ribosomal, respiratory and photosynthetic machineries, RNA editing in plant mitochondria and chloroplasts has significant organismal consequences. The absence of RNA editing, e.g., due to mutations in PPR proteins, can lead to severe phenotypes including sterility, growth retardation, abnormal embryo and endosperm development, as well as embryonic lethality [7].

Analyses of the editing sites of mitochondrially-encoded subunits have shown that the edits generally increase the hydrophobicity of the target protein, and that they generally restore the identity of the amino acid to the evolutionarily conserved residue in non-edited plant species [6,8]. Many edits also remove prolines from the protein sequence. While the impact of RNA editing is clear at the sequence and phenotypic levels, the underlying biochemical and molecular explanations for these effects have remained obscure, particularly in the absence of three-dimensional structures of the edited proteins. Recently, however, high-resolution structures of complexes of the plant respiratory chain, complex I, complex III_2_, and complex IV have been obtained [9,10,11,12].

Respiration occurs through an electron transport chain in the inner mitochondrial membrane composed of four membrane-embedded multi-subunit enzymes (complexes I–IV) and two electron carriers (quinone and cytochrome *c*). The complexes sequentially transfer electrons from reduced metabolites (NADH and succinate) to molecular oxygen while concomitantly pumping protons against their concentration gradient. This generates a proton electrochemical potential that is dissipated by ATP synthase (also called complex V), producing ATP [13] (Figure 1A). Electron transfer through complexes I-IV is enabled by essential co-factors such as hemes, copper centers and iron–sulfur clusters. In plants, respiration is coordinated with photosynthesis to produce energy and metabolic intermediates for the organism and to maintain the redox balance of the cell [14]. Respiratory complexes are also involved in hormonal, defense and stress responses [15,16]. Thus, disruption of respiration can lead to increases in reactive oxygen species, an upregulation of “alternative” (non-proton pumping) respiratory pathways and overall stunted growth and development, as seen in most PPR mutants.

In this perspective article, we took advantage of the recently published structures of plant respiratory complexes [11,12] to close the disparity between our phenotypic and biochemical understanding of RNA editing in plant mitochondria. We examined the RNA-editing sites in the coding regions of the edited subunits of plant respiratory complex I, complex III_2_ and complex IV from a structural perspective. We produced an extensive list of known RNA edits in land plants and analyzed the structural impact of the amino acid change in terms of its biochemical features and location in the protein. We also integrated this characterization with sequence alignments of respiratory chain protein subunits between plants and other prokaryotic and eukaryotic model organisms. Through examples of single- and multi-target PPR proteins, we demonstrate how we can use this information to explain the deleterious phenotypes of PPR mutants. This work shows how the structural perspective can bridge the gap between sequence and phenotype and provides a framework for the continued analysis of future PPR mutants.

## 2. Approach

We structurally characterized the RNA-editing sites of mitochondrially encoded subunits of plant respiratory-chain complexes for which there are high-resolution structures, i.e., complex I, complex III_2_ and complex IV. These subunits are NAD1, NAD2, NAD3, NAD4, NAD4L, NAD5, NAD6, NAD7, NAD9 in complex I, COB in complex III_2_, and COX1, COX2 and COX3 in complex IV (Figure 1B–D). We first used standardized literature searches to create a near-comprehensive list of experimentally identified sites across land plants, determining both the genomically encoded and the edited residues. We then examined the degree of conservation of the edited residues in plants and in a broad range of prokaryotic and eukaryotic model organisms used in respiratory chain research, as well as in humans (*Paracoccus denitrificans*, *Saccharomyces cerevisiae*, *Yarrowia lipolytica*, *Drosophila melagonaster*, *Homo sapiens*). Next, we mapped each edit to the available structures of complex I (*Arabidopsis thaliana*, PDB: 7AR8) [12], complex III_2_ (*Vigna radiata*, PDB: 7JRG) [11] and complex IV (*V. radiata*, PDB: 7JRO) [11]. We characterized the structural and biochemical features of each edit by assessing (1) location within the mitochondrial membrane, (2) location in the vicinity of functional residues (e.g., complex I’s quinone binding site or hydrophilic axis, complex III_2_′s hemes, complex IV’s hemes and putative proton channels), (3) location at an interface with other subunits, (4) change in the biochemical properties of the amino acid (e.g., hydrophilic to hydrophobic), and (5) location relative to secondary structure elements (i.e., removal of a proline from an α-helix or a β-strand). All of these aspects have structural and functional implications. Lastly, we identified the editing sites that have been linked to detrimental phenotypes in mutant screens.

### 2.1. Structural and Sequence Characterization of Editing Sites

Our approach identified 206 sites in complex I, 21 in complex III_2_ and 48 in complex IV across 17 plant species. Of these, 65% to 80% are located within the transmembrane region of the proteins, 45% to 65% are edited from hydrophilic to hydrophobic residues, 15% to 25% remove prolines in α-helices or β-sheets, 15% to 40% are at interfaces between subunits and 10% to 50% are close to functionally important residues (Table 1 and Appendix A). Notably, a third of the edits in NAD7 (complex I), COB’s (complex III_2_) and half of the edits in COX1 (complex IV) are in the vicinity of key functional residues, such as those involved with quinone binding sites, heme coordination or proton-translocation channels [17,18,19]. Additionally, 20% to 50% of the edits restored residues to highly conserved amino acids, defined as being invariant across all our analyzed plant and non-plant species, from bacteria (*P. denitrificans*) to humans. In NAD7, COX1 and COX3 up to 67% of the edited sites occur at (and restore) highly conserved residues. Overall, only 6 of the 275 total editing sites (~2%) lack any salient structural characteristics (e.g., a non-conserved residue that is not in the membrane or at an interface or close to functional residues). Thus, our analysis revealed that almost all the editing sites in plant complex I, complex III_2_ and complex IV have implications for the structure, and therefore the function, of the respiratory complexes.

To further assess the functional impact of these editing sites, we examined the existing literature on mutants lacking RNA editing, e.g., due to mutations in PPR proteins. In the next section, we discuss several of these mutants and show how their phenotypic effects can be explained from a structural perspective.

### 2.2. Structural and Functional Consequences of PPR Mutants

#### 2.2.1. Mutations in Single-Site PPR Proteins

PPR proteins use protein: RNA interactions to recognize patterns in the mRNA and position the deamination active site. Depending on their cognate pattern, PPRs can have single or multiple targets across the same or different subunits in the same or different complexes. Mutants of single-site PPRs are an excellent tool to link the structural impact of an RNA edit to its functional relevance and illustrate the power of taking a structural perspective. Here, we discuss the effects of mutations in *MPR25* [20], *SMK1* [21], *PpPPR_79* [22], *DEK605* [23] and *MEF9* [24] (Figure 2). In line with the structures used, residue numbers in complex I refer to the protein sequences of *A. thaliana* subunits, while numbers in complex III_2_ and complex IV refer to the protein sequences in *V. radiata* subunits.

MPR25 edits NAD5′s serine-527 to leucine (Figure 2B). NAD5 is a transmembrane subunit of complex I’s membrane arm with essential roles in the complex’s proton pumping mechanism. The *NAD5*-S527L editing site is conserved in several plant species. Additionally, this leucine is conserved in *P. denitrificans*, *Y. lipolytica* and *H. sapiens*. *MPR25* mutants show growth retardation, pale green leaves, a reduced number of tillers and an upregulation of the alternative respiratory chain [20]. Structurally, the S527L edit replaces a hydrophilic residue that is exposed to the membrane with a hydrophobic residue. Due to the unfavorable energetics of burying a hydrophilic residue within the hydrophobic lipid environment of the membrane, it is well understood that hydrophilic residues destabilize transmembrane helices and can block proper membrane insertion [25]. Therefore, the presence of serine as NAD5′s residue 527 would lead to a less stable subunit that would be more difficult to insert into the membrane to assemble complex I, leading to the *MPR25* mutant phenotype due to reduced complex I assembly.

Another single-site PPR acting on NAD5 is PpPPR_79, which edits arginine-200 to cysteine (Figure 2C). The cysteine is conserved in *D. melanogaster* and *H. sapiens*, and the editing site is present in *A. thaliana* and *O. berteroana*. *PpPPR_79* mutants show severe growth retardation [22]. The *NAD5*-R200C edit replaces a large residue with a much smaller one at the interface with complex I’s subunit NDUFB10. In mammals, NDUFB10 is essential for the assembly and integrity of complex I by bridging NAD5 with another essential proton-pumping subunit, NAD4 [26,27]. The packed interface between NAD5 and NDUFB10 would not be able to accommodate a large sidechain like that of arginine. Therefore, the lack of editing at this site would disrupt the interaction between NAD5 and NUDFB10. This would lead to a weakening and potential disruption of complex I’s membrane arm, likely resulting in reduced complex I assembly and activity.

PPR proteins also edit other essential complex I transmembrane subunits involved in proton pumping, such as NAD1. For instance, DEK605 edits *NAD1′*s serine-203 to phenylalanine. The *NAD1*-S203F edit is found in *A. thaliana* and maize, and the phenylalanine is also conserved in all the non-plant organisms examined. *DEK605* mutants show reduced complex I levels and activity, up-regulated alternative respiratory pathways and disrupted embryonic and seed development, with less than 20% of seeds germinating into weak seedlings [23]. Both hydrophobicity and location effects explain the phenotype in *DEK605* mutants. NAD1′s phenylalanine-203 is exposed to the membrane; thus, the edit replaces a hydrophilic residue with a more favorable, hydrophobic one. More importantly, phenylalanine-203 is in a loop that forms the “Q tunnel” through which complex I’s substrate quinone accesses its binding site (Figure 2D). Thus, a lack of editing in this position would be expected to have significance functional consequences, as seen in the mutants.

*NAD7* is also subject to multiple single-site PPR proteins. NAD7 is a core subunit in complex I’s matrix arm that contains the quinone reduction site and is thus essential to complex I activity. PPR protein SMK1 edits *NAD7′*s proline-279 to leucine (Figure 2E). The *NAD7*-P279L site is conserved in various plants and the edited leucine is conserved in *D. melanogaster* and *H. sapiens*. *SMK1* mutants have a severe phenotype: their reduced complex I levels and activity, abnormal mitochondrial biogenesis and abnormal embryo/endosperm development lead to embryonic or seedling lethality [21]. Leucine-279 is located near the C-terminus of a conserved α-helix between residues 258–281. The loop that follows this helix is a structural motif that allows NAD7 to bind to NDUFS6, NDUFS1 and NDUFS8 (Figure 2E). These are important interactions at the interface between the “modules” that compose complex I’s peripheral arm. Whereas NAD7 and NDUFS8 belong to complex I’s Q module, which contains the quinone binding site, NDUFS1 and NDUFS6 belong to the N module, which contains the NADH binding site. In plants, the joining of the Q module to the N module is one of the first steps in complex I’s assembly pathway [28]. Without a proper N/Q assembly, the whole assembly process of plant complex I is abrogated. The presence of a proline at NAD7-279 would destabilize the C-terminal end of its helix, thereby changing the conformation of the subsequent loop and disrupting the N/Q interface. Therefore, the lack of editing at *NAD7*-P279L would lead to an inability to properly assemble complex I, explaining the severe phenotype of *SMK1* mutants.

A structural perspective also sheds light on the mild phenotypes seen for some PPR mutants such as MEF9, which edits *NAD7′*s serine-67 to phenylalanine (Figure 2F). This editing site is present in *A. thaliana*, *V. radiata* and *Orya sativa*, and phenylalanine-67 is conserved in *Y. lipolytica*, *D. melanogaster* and *H. sapiens*. *MEF9-1* mutants show a normal development and growth phenotype [24]. Although this result was surprising to the original researchers, it can be rationalized structurally. The *NAD7*-S67F edit changes a small polar residue to a large hydrophobic one. Given that the site is in a short helix in the matrix, a change to a hydrophobic residue would not add a significant advantage; additionally, given that serine is smaller than phenylalanine, producing potential steric clashes in this region upon the lack of editing is not a concern.

The structural approach is also fruitful to explain other phenotypes that previously seemed surprising. For instance, *SMK4* mutants fail to edit COX1’s proline-497 to serine, leading to slow growth and development, small plants, delayed flowering and small kernels, as well as reduced complex IV levels and an upregulation of the alternative respiratory chain [29]. The serine is conserved in *S. cerevisiae*, *H. sapiens* and several plant species. Using homology modeling with the structure of *Rhodobacter sphaeroides* complex IV, the authors identified that proline-497 is in a matrix loop rather than in a helix and were surprised by the strong phenotype given this peripheral location. The structure of *V. radiata* complex IV allows us to understand the biochemical basis of this edit and resolve this “mystery” (Figure 2G–H). Despite not being in a helix, COX1-P497S has significant structural relevance. Together with COX1-asparigine-496, COX1-serine-497 forms a hydrogen bond with an adjacent β-strand on COX5B (yeast COX4), effectively extending its β-sheet. The COX5B 113-119 β-strand is immediately downstream of COX5B-His-110, which is one of four Zn^2+^-binding residues in this subunit (Figure 2H). The Zn^2+^ binding in COX5B is essential for complex IV assembly and stability [30]. If COX1-497 remained a proline, no hydrogen bond could be formed with COX5B’s β-sheet. This would likely lead to an incorrect positioning of the sheet, which would consequently misplace COX5B’s essential hisitidine-110 residue, distorting the geometry and reducing its ability to coordinate the Zn^2+^ ion.

#### 2.2.2. Mutations in Multi-Site PPR Proteins

Many PPR proteins edit multiple subunits across different mitochondrial complexes. As these mutations can simultaneously affect respiration, heme biogenesis, mitochondrial translation and solute transport, their phenotypic effects are generally severe. Given that not all these PPR targets have structures available, we illustrate our analysis with two mutants that exert their effects through the respiratory complexes for which there are structures (Figure 3).

PpPPR_77 edits two sites in complex IV’s core subunits COX2 and COX3. The *COX3*-R245W and *COX2-*R122W edits restore tryptophan residues that are highly conserved across prokaryotes and eukaryotes. The lack of editing in *PpPPR_77* mutants results in severe growth retardation in moss [31]. This is readily understandable from the structures (Figure 3A-C). The *COX3*-R245W edit replaces a positive amino acid in the hydrophobic core of the subunit with a much more hydrophobic side chain (Figure 3B). The *COX2*-R122W edit is even more significant. COX2 forms the binding site for cytochrome *c* (cyt *c*) and contains a di-copper redox center (Cu_A_). This center accepts the electron from cyt *c* and transfers it to the hemes and binuclear center in COX1, where the last step of respiration occurs by reducing O_2_ to water. Cu_A_ is coordinated by two histidines, one cysteine and one methionine. If these residues were incorrectly positioned or if the electrostatic environment of the redox center were altered, binding of cyt *c* and electron transfer would be jeopardized, precluding electron transfer through the whole canonical respiratory chain. COX2-W122 is in the immediate vicinity of the copper-coordinating residues and key cyt *c* binding residues (Figure 3C). The presence of a positively charged arginine here would very likely have a strong impact on the redox state of Cu_A_ and its electron-transfer abilities. Moreover, in plants, bacteria, metazoans and fungi, cyt *c* binds to COX2 through electrostatic interactions: cyt *c* presents a positively charged surface and COX2 a negatively charged one, with tyrosine-123 being a key interacting residue [32]. The presence of a positive charge in position 122 would also very likely lead to disrupted cyt *c* binding. Therefore, the lack of editing at COX2-R122W would be doubly deleterious, affecting both cyt *c* binding and the redox poise of Cu_A_.

A similar effect is in place for *DEK10* mutants, which fail to edit *COX2*-P179S and *NAD3*-P22L (Figure 3D-F), leading to significant reductions in complex I and complex IV activity and levels and a classic defective-kernel phenotype in maize [33]. COX2′s serine-179 is towards the end of a β-strand that leads to one of the four Cu_A_-coordinating residues (histidine-184) (Figure 3D). The lack of editing of *COX2*-P179S would leave a proline, which would remove hydrogen bonding potential with two residues on the loop that positions the essential histidine-184 (Figure 3D). This would lead to a de-structuring of the loop and the likely mis-localization of histidine-184. Similarly, the lack of editing at *NAD3*-P22L would cause a premature end to a helix that leads to a key loop of complex I’s quinone-binding tunnel (Figure 3E–F), deleteriously affecting complex I’s enzymatic activity.

## 3. Conclusions 

We used the recently published cryoEM structures of plant respiratory complex I, complex III_2_ and complex IV [9,10,11,12] to analyze the impact of mutations in RNA-editing enzymes. By placing the edits in their structural context, we determined that most of them have readily available structural and biochemical explanations for their functional effects. Moreover, as previously discussed at the molecular and cellular levels [7], the functional effects of the edits can be distilled to a few structural and biochemical themes and principles. Most edits affect residues in the membrane, which explains the benefits of replacing polar with more hydrophobic amino acids. Many edits also remove otherwise disruptive prolines from key secondary-structure motifs. While the effects of removing prolines from presumed α-helices had been proposed by some PPR researchers, the analogous effect on β-sheets had not been noted. Furthermore, several edits occur at subunit interfaces that would be disrupted by residues of a different charge or size. Finally, many edited residues are in the vicinity of functionally essential residues of respiratory-complex subunits, be they substrate binding sites, proton conductance pathways or electron transfer centers.

Our work demonstrates that a structural analysis of RNA-editing sites can provide insight into the functional and physiological effects of RNA editing. Moreover, it provides a framework for the future discussion of PPR phenotypes in plant mitochondria as well as in chloroplasts, where we expect that the same general biochemical and structural principles are in effect. However, not all structures of edited proteins are currently available. We predict that structures of the remaining proteins that are edited will reveal shared principles for the biochemical basis of RNA-editing mutants in plant organelles.

## Figures and Tables

**Figure 1 ijms-23-00684-f001:**
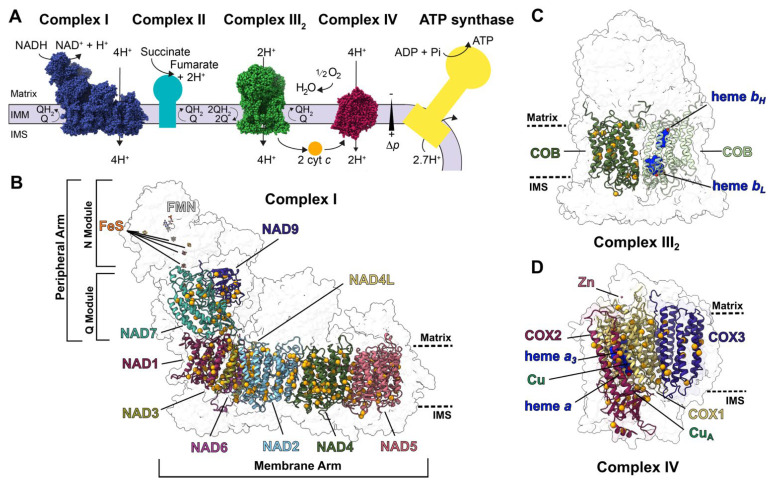
RNA-edited subunits of plant respiratory complexes: (**A**) Schematic overview of the plant canonical respiratory chain. Complexes I-IV and ATP synthase are shown in the inner mitochondrial membrane (IMM). Complex I (blue, PDB: 7AR8), complex III_2_ (green, PDB: 7JRG) and complex IV (magenta, PDB: 7JRO) are shown with their atomic structures in sphere representation [11,12]. Complexes without structures (complex II, teal; ATP synthase, yellow) are represented in boxes. Substrates and products (NADH, NAD^+^, succinate, fumarate, O_2_, H_2_O, ADP, P_i_, ATP), electron carriers (quinone, Q; quinol, QH2; cytochrome *c*, cyt *c*), proton pumping stoichiometry (H^+^), protonmotive force (∆p), matrix, intermembrane space (IMS) and inner mitochondrial membrane (IMM) are indicated. (**B**–**D**) Subunits of complex I (**B**), complex III_2_ (**C**) and complex IV (**D**) that undergo RNA editing are shown in colored cartoons. Edited residues are shown as orange spheres overlayed over the transparent surface of the complex. Approximate locations of the matrix and IMS are shown in dashed lines. (**B**) In addition to the edited subunits, complex I’s flavin mononucleotide (FMN) and iron-sulfur (FeS) co-factors are shown in stick representation. Complex I’s membrane and peripheral arms and approximate locations of the N (NADH-binding) and Q (quinone-binding) modules are marked. (**C**) Both COB subunits of complex III_2_ are shown. Dark-green COB highlights the edited residues. Light-green COB highlights the subunit’s heme *b*_H_ and *b*_L_ (dark blue spheres). (**D**) Complex IV’s heme *a*, heme *a*_3_, Zn^2+^, copper-A (CuA) co-factors are shown in spheres.

**Figure 2 ijms-23-00684-f002:**
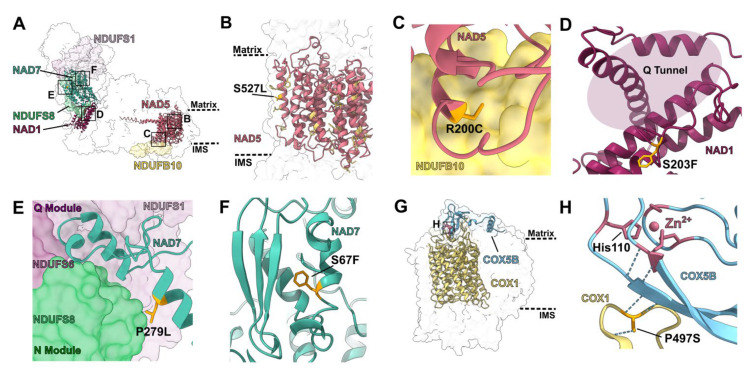
Structural characterization of edits by select single-site PPR proteins: (**A**) Overall position of the edits produced by PPR proteins MPR25, PpPPR_79, SMK1 and MEF9 in *A. thaliana* complex I (PDB: 7AR8). Edited subunits are shown in colored cartoon and interacting subunits are shown in colored surface over transparent complex I surface. Insets show the positions of panels (**B**–**F**). Approximate locations of the matrix and IMS are shown in dashed lines. (**B**–**F**) Structural details of edited residues. Edited residues are shown in orange stick representation. (**B**) NAD5′s transmembrane region (pink). NAD5′s residue 527, located in the inner mitochondrial membrane, is edited by MPR25 from a hydrophilic serine to a hydrophobic leucine (S527L). Other edited residues in the transmembrane region of NAD5 are colored in light yellow. (**C**) Interface between NAD5 (pink cartoon) and NDUFB10 (yellow surface). NAD5 residue 200 is edited by PpPPR_79 from arginine to cysteine (R200C). Editing of a bulky to a small residue allows this helix to fit in the crowded interface. (**D**) NAD1′s residue 203, located in a loop close to the quinone tunnel, is edited from serine to phenylalanine (S203F) by DEK605. Editing to a more hydrophobic residue improves the stability of this buried loop and its ability to bind quinone. (**E**) Interface between NAD7 (teal cartoon, Q module) with NDUFS8 (light green surface, Q module), NDUFS6 (dark purple surface, N module), NDUFS1 (light purple surface, N module). NAD7′s residue 279 is edited by SMK1 from proline to leucine (P279L). Destabilization of the helix of L279 would impair the interaction between the N and Q modules of complex I. (**F**) NAD7′s residue 67 is edited from serine to phenylalanine (S67F) by MEF9. Given that this position is not in the membrane, the change to a hydrophobic residue does not have a significant advantage. (**G**) Overall position of the edit produced by SMK4 in *V. radiata* complex IV (PDB: 7JRO). Edited subunit (COX1) is shown in yellow cartoon and interacting subunit (COX5B) is shown in light blue cartoon over transparent complex IV surface. Insets show the positions of panels (**H**). Approximate locations of the matrix and IMS are shown in dashed lines. (**H**) COX1′s residue 497 is edited by SMK4 from a proline to a serine (P497S). Serine-497 forms multiple hydrogen bonds (dashed light blue lines) that help position the Zn^2+^-coordinating residues of COX5B (pink sticks, pink atom).

**Figure 3 ijms-23-00684-f003:**
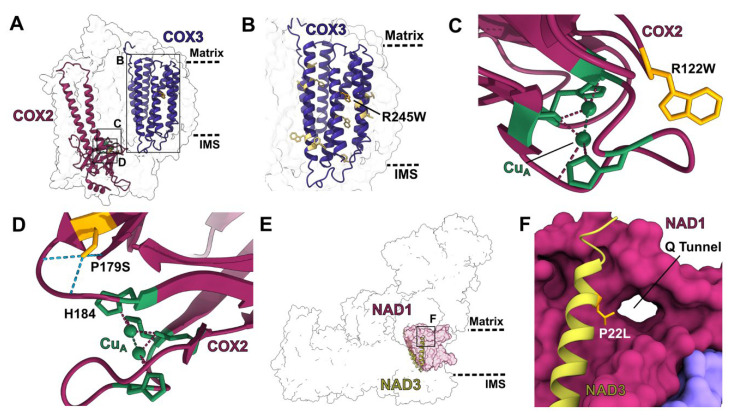
Structural characterization of edits by select multi-site PPR proteins: (**A**) Overall position of the edits produced by PPR protein PpPPR_77 in *V. radiata* complex IV (PDB: 7JRO). Edited subunits are shown in colored cartoon. Insets show the positions of panels (**B**–**D**). (**B**–**D**) Structural details of edited residues. Edited residues are shown in orange stick representation. (**B**) COX3′s transmembrane region (dark purple cartoon). COX3′s residue 245 is edited from arginine to tryptophan (R245W), removing a positively charged residue from the hydrophobic environment of the membrane. Other edited residues in COX3′s transmembrane region are shown in light yellow stick. Approximate location of the matrix and IMS are shown in dashed lines. (**C**) Copper site for electron transfer from cytochrome *c* (not shown) in COX2 (maroon cartoon). Copper atoms of Cu_A_ center are shown in green, with coordinating residues in green stick and coordination bonds in dashed lines. COX2′s residue 122 is edited from arginine to tryptophan (R122W). The positive charge of the arginine would alter the electronic environment of the Cu_A_ center. (**D**–**F**) Structural details of edits produced by PPR protein DEK10 in *V. radiata* complex IV and *A. thaliana* complex I. (**D**) Cu_A_ site of COX2 as in panel (**C**). DEK10 edits COX2′s residue 179 from proline to serine (P179S). Serine-19 forms multiple hydrogen bonds (light blue dashed lines) that position the loop that leads to Cu_A_-coordinating residue histidine-184 (H184). (**E**) Overall position of the DEK10 edit in *A. thaliana* complex I (PDB: 7AR8). Edited subunit (NAD3) is shown in light green cartoon. Interaction subunit (NAD1) is shown in magenta surface. Inset shows the position of panel (**F**). (**F**) NAD3′s interface with NAD1 at the entrance of the Q tunnel, through which complex I’s substrate quinone enters its active site. DEK10 edits NAD3′s residue 22 from proline to leucine (P22L). A proline in this position would break the helix, affecting the access of quinone to the active site.

**Table 1 ijms-23-00684-t001:** Summary of the characteristics of RNA editing positions in plant respiratory complex I, complex III_2_ and complex IV. RNA-editing positions were collated from 17 plant species. Structural characteristics were assessed based on the homologous position of the edit in the high-resolution structures of *A. thaliana* (complex I, PDB: 7AR8) or *V. radiata* (complex III_2_, PDB: 7JRG; complex IV, PDB: 7GRO). Full details available in Appendix A. CI, complex I; CIV, complex IV; MA, membrane arm; PA, peripheral arm.

		Conservation			Structural Type					
Subunit	# Edits	High	Inter-Mediate	Low/None	In Membrane	To Hydrophobic	Proline Removal	Close to Functional	At Interface	Unaccounted
Complex I										
NAD1	21	29%	43%	29%	86%	48%	10%	0%	33%	0%
NAD2	36	0%	11%	89%	100%	61%	19%	14%	31%	6%
NAD3	19	16%	47%	37%	89%	47%	32%	11%	74%	0%
NAD4	38	16%	24%	61%	100%	50%	18%	11%	34%	0%
NAD4L	14	0%	36%	64%	100%	79%	21%	21%	86%	0%
NAD5	32	16%	16%	69%	91%	63%	9%	3%	28%	3%
NAD6	11	0%	18%	82%	100%	45%	36%	18%	73%	0%
Total MA	171	12%	25%	63%	95%	56%	19%	10%	43%	2%
NAD9	13	38%	31%	31%	0%	69%	0%	0%	38%	0%
NAD7	22	50%	36%	14%	0%	82%	5%	32%	23%	0%
Total PA	25	46%	34%	20%	0%	77%	3%	20%	29%	0%
Total CI	206	17%	27%	56%	79%	60%	16%	12%	41%	1%
Complex III_2_										
COB	21	29%	29%	43%	90%	43%	24%	33%	14%	10%
Complex IV										
COX1	15	53%	33%	13%	93%	53%	7%	53%	13%	0%
COX2	21	33%	38%	29%	24%	71%	10%	38%	29%	5%
COX3	12	67%	33%	0%	100%	67%	42%	17%	50%	0%
Total CIV	48	48%	35%	17%	65%	65%	17%	38%	29%	2%

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
