# Peer review of "A Structural Perspective on the RNA Editing of Plant Respiratory Complexes"

_ijms, 2022, doi:10.3390/ijms23020684_

Round 1
Reviewer 1 Report
This Perspective paper is very interesting to demonstrate how the structural information of respiratory complexes can explain the phenotypes of RNA-editing mutants. Based on the recently determined structures of plant respiratory complex I, complex III2, and complex IV, as well as the impact of the amino acid changes of RNA editing in terms of their location and biochemical features, they nicely showed that the structural perspective can bridge the gap between sequence and phenotype observed in the RNA-editing mutants in plant mitochondria. Few minor corrections are needed.
1. In the last paragraph of Abstract, it is described that this work shows how the structural perspective can bridge the gap between sequence and phenotype and provides a framework for the continued analysis of RNA-editing mutants in plant mitochondria and chloroplasts, which is overstatement. The current paper deals with only mitochondrial respiratory complexes but not chloroplasts.
2. Figure 1A; inner mitochondrial membrane (IMM) should be labeled on Figure 1A, and matrix and intermembrane space (IMS) should be indicated on Figure 1A.
3. Figure 2; in Figure 2G legend, it is described that approximate location of the matrix and IMM are shown in dashed lines. However, no dashed lines are shown on Figure 2G. In addition, IMS was labeled on the figure but IMM was mentioned in figure legend. This discrepancy should be corrected.
4. Figure 3; IMS was labeled on the figure but IMM was mentioned in figure legend. This discrepancy should be corrected.
5. Line 33; pentatricopeptide (PPR) proteins should be pentatricopeptide repeat (PPR) proteins.
6. Line 113, 114, 120, 197, 229, 230, 234, 293, 333, 334; a-helices and b-sheet should be alpha (or α)-helices and beta (or β)-sheet.
Author Response
We thank the reviewer for his/her comments and corrections. We address each point below.
- In the last paragraph of Abstract, it is described that this work shows how the structural perspective can bridge the gap between sequence and phenotype and provides a framework for the continued analysis of RNA-editing mutants in plant mitochondria and chloroplasts, which is overstatement. The current paper deals with only mitochondrial respiratory complexes but not chloroplasts.
- We meant to say that the general biochemical and structural principles learned from the RNA edits in mitochondria will most likely also be applicable as explanatory principles for RNA editing in chloroplasts. We have re-worded this more clearly in the abstract and last paragraph of the discussion.
- Figure 1A; inner mitochondrial membrane (IMM) should be labeled on Figure 1A, and matrix and intermembrane space (IMS) should be indicated on Figure 1A.
- We apologize for this oversight. This has been corrected.
- Figure 2; in Figure 2G legend, it is described that approximate location of the matrix and IMM are shown in dashed lines. However, no dashed lines are shown on Figure 2G. In addition, IMS was labeled on the figure but IMM was mentioned in figure legend. This discrepancy should be corrected.
- We apologize for this oversight. This has been corrected.
- Figure 3; IMS was labeled on the figure but IMM was mentioned in figure legend. This discrepancy should be corrected.
- We apologize for this oversight. This has been corrected.
- Line 33; pentatricopeptide (PPR) proteins should be pentatricopeptide repeat (PPR) proteins.
- We apologize for this oversight. This has been corrected.
- Line 113, 114, 120, 197, 229, 230, 234, 293, 333, 334; a-helices and b-sheet should be alpha (or α)-helices and beta (or β)-sheet.
- We apologize for this formatting error, and thank the reviewer for pointing out all the lines. The alpha and beta symbols were lost when the font was changed from Calibri/Symbol to Arial. This has been corrected throughout.
Reviewer 2 Report
The review entitled " A structural perspective on the RNA editing of plant respiratory complexes" provide a very nice perspective of RNA editing in the proteins involved in the respiratory complexes plants and their functional consequences. The article is beautifully written and explained and can be accepted for publication after few minor edits:
1) For all the protein structures described in the article please provide the PDB ID of the same.
2) Provide the full name of the abbreviation used.
3) Instead of using a-helix and b-sheet I recommend using the greek letters: α-helix and β-sheet. Otherwise the use of a-helix and b-sheet may confuse people who aren't familiar with the structural biology terminology.
Author Response
We thank the reviewer for his/her kind comments and the corrections. We respond to each point below.
1) For all the protein structures described in the article please provide the PDB ID of the same.
- PDB IDs have been added to line 108 and the legend of Fig 1.
2) Provide the full name of the abbreviation used.
- We are unsure what abbreviation the reviewer is referring to. We have removed the abbreviations of CI, CIII2 and CIV from line 99 and added them in line 49, if that is what the reviewer was referring to.
3) Instead of using a-helix and b-sheet I recommend using the greek letters: α-helix and β-sheet. Otherwise the use of a-helix and b-sheet may confuse people who aren't familiar with the structural biology terminology.
- We agree. This was a formatting error. We have updated all alpha and beta with Greek letters.